# Perceived and mentally rotated contents are differentially represented in cortical depth of V1

Polina Iamshchinina [1,2 ✉], Daniel Kaiser [3], Renat Yakupov [4], Daniel Haenelt [5], Alessandro Sciarra[6,7], Hendrik Mattern[6], Falk Luesebrink[6,7], Emrah Duezel[4,8], Oliver Speck[4,6,8,9], Nikolaus Weiskopf [5,10] & Radoslaw Martin Cichy[1,2]

Primary visual cortex (V1) in humans is known to represent both veridically perceived external input and internally-generated contents underlying imagery and mental rotation. However, it is unknown how the brain keeps these contents separate thus avoiding a mixture of the perceived and the imagined which could lead to potentially detrimental consequences. Inspired by neuroanatomical studies showing that feedforward and feedback connections in V1 terminate in different cortical layers, we hypothesized that this anatomical compartmentalization underlies functional segregation of external and internally-generated visual contents, respectively. We used high-resolution layer-specific fMRI to test this hypothesis in a mental rotation task. We found that rotated contents were predominant at outer cortical depth bins (i.e. superficial and deep). At the same time perceived contents were represented stronger at the middle cortical bin. These results identify how through cortical depth compartmentalization V1 functionally segregates rather than confuses external from internally-generated visual contents. These results indicate that feedforward and feedback manifest in distinct subdivisions of the early visual cortex, thereby reflecting a general strategy for implementing multiple cognitive functions within a single brain region.

[1] Department of Education and Psychology, Freie Universität Berlin, Berlin, Germany. [2] Berlin School of Mind and Brain, Humboldt-Universität zu Berlin, Berlin, Germany. [3] Department of Psychology, University of York, Heslington, York, UK. [4] German Center for Neurodegenerative Diseases (DZNE), Magdeburg, Germany. [5] Department of Neurophysics, Max Planck Institute for Human Cognitive and Brain Sciences, Leipzig, Germany. [6] Department of Biomedical Magnetic Resonance, Institute for Physics, Otto-von-Guericke-University, Magdeburg, Germany. [7] Department of Neurology, Otto-von-Guericke University, Magdeburg, Germany. [8] Center for Behavioral Brain Sciences, Magdeburg, Germany. [9] Leibniz Institute for Neurobiology, Magdeburg, Germany. [10] Felix Bloch Institute for Solid State Physics, Faculty of Physics and Earth Sciences, Leipzig University, Leipzig, Germany. ✉email: iamshchinina@gmail.com

Mental rotation is at the core of efficiently acting upon objects regardless of their orientation, such as when searching for a nail in a toolbox or solving a Rubik's cube. It comprises the perception of an external input and the internal generation of a transformed representation[1–3]. Recent studies demonstrated that both operations are concurrently mediated by primary visual cortex[4,5] (V1). Given this spatial overlap, how does the brain separate perception and mental rotation? Why do we not confuse perceived and mentally transformed contents?

Recent neuroanatomical studies suggest that projections carrying external and internally generated signals in V1 are segregated across cortical layers. Feedforward projections terminate in the middle layer, while feedback connections terminate in superficial and deep layers[6–11]. Studies of working memory and attention demonstrated the functional relevance of this layer-specific separation[12–19]. However, these studies measured the retention or amplification of the very stimuli previously represented in V1 or estimated perception signal not concurrently but in a separate task. It is thus unclear how V1 separates presented from internally modified contents, such as during mental rotation.

Here, using high-resolution fMRI at 7 T we show that the concurrent representation of perceived and mentally transformed contents during mental rotation is enabled by cortical depth separation of information in V1. The perceived contents were strongest at the middle cortical depth bins, while mentally rotated contents dominated in the superficial and deep cortical bins. These results show how the perceived and mentally rotated contents are mediated by functionally distinct neural representations, explain why externally induced and internally generated contents are not confused, and supports the view of V1 as a dynamic 'blackboard' updated through connections from higher-order areas rather than a low-level stage of hierarchical processing.

## Results

We recorded 7 T fMRI with 0.8 mm iso voxel resolution while participants ($N = 23$) viewed and mentally rotated oriented gratings[4]. On each trial, we presented a single grating (15°, 75°, or 135°), followed by a cue that instructed participants to mentally rotate the presented grating to the left or to the right for either 60° or 120° (Fig. 1A). Thus, each of the orientations presented in the trial could be turned into one of the two other orientations (Fig. 1B). The presented and rotated gratings were different from each other on every trial, allowing us to independently assess encoding of perceived and mentally rotated contents. At the end of each trial, participants compared their mental rotation result to probe grating with similar orientation. Behavioural data confirmed that the participants could successfully perform this task, with greater reaction times ($t_{23} = 3.4$, $p = 0.0012$) and error rates ($z = -1.64$, $p = 0.06$) for the larger rotation angle[1]. (Supplementary Fig. 1A).

To examine depth-specific responses during mental rotation and perception, we extracted three gray matter depth bins approximating deep, middle, and superficial cortical layers in V1 in every participant (Fig. 2A, see Methods for further clarification). For each depth bin, we trained support vector machine classifiers to differentiate multi-voxel response patterns evoked by the three grating orientations. Classifiers were always trained on response patterns in a separate block-design localizer, during which participants saw the three orientations while performing an orthogonal task (see Methods). These response patterns served as a benchmark for a strong orientation-selective response in V1. Classifiers were then tested on response patterns in the mental rotation experiment. To investigate representations of perceived and rotated contents across time, we performed separate classification analyses for every timepoint from stimulus onset to the end of the trial (5 TRs in total). We then analysed the predicted orientation (15°, 75°, or 135°) at each timepoint (Fig. 1C). To estimate the representational strength of the presented and rotated grating orientations, we counted how often classifiers predicted (1) the presented orientation (e.g. predicting 15° on a trial where a 15° grating was rotated into a 75° grating), (2) the rotated orientation (e.g. predicting 75° on a trial where a 15° grating was rotated into a 75° grating), and (3) the third, unused orientation (e.g. predicting 135° on a trial where a 15° grating was rotated into a 75° grating). Accumulating the classifier predictions across trials, we were able to track representations of perceived and mentally rotated contents across cortical depths (see Methods).

We performed in-depth analyses in the time interval from 8 to 10 seconds (i.e. 2 TRs) after the rotation cue. This time interval

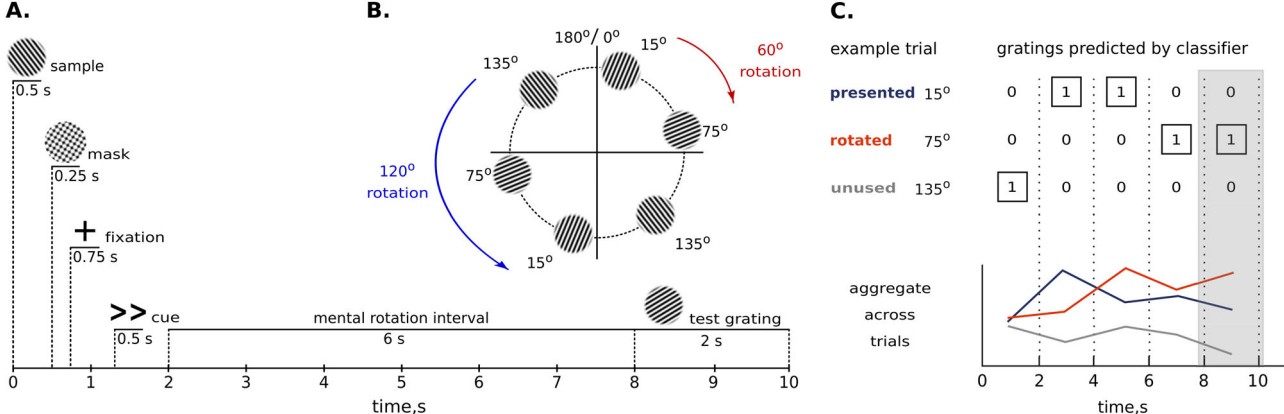

**Fig. 1 Experimental methods. A** On each trial, participants viewed a sample grating and then had 6 seconds to rotate it 60° (<, >) or 120° (≪, ≫) to the left or to the right. After the mental rotation interval, participants had 2 seconds to report whether a probe grating was tilted clockwise or counterclockwise compared to the mentally rotated grating. **B** We used a set of three stimuli, 15°, 75°, and 135° oriented gratings. As a result of the mental rotation, each stimulus could be turned into one of the other two stimuli. For example, rotation of a 15° grating (red arrow) for 60° clockwise results in a 75° grating or rotation of a 135° grating (blue arrow) 120° counterclockwise results in a 15° grating. **C** This panel shows classifiers' decisions in an example trial, in which a 15° grating was rotated into a 75° grating. We aggregated results across trials by counting how often classifiers predicted the presented orientation, the rotated orientation, and the unused orientation. The shaded area denotes the time interval chosen for the in-depth analysis (measurements at 8 and 10 seconds).

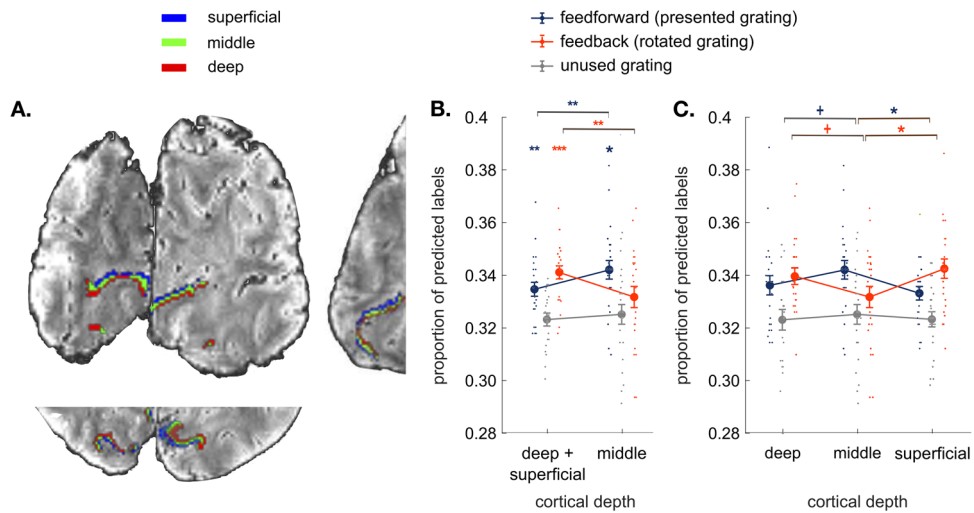

**Fig. 2 Results. A** Coronal, axial, and sagittal slices of the average EPI image of a representative participant, overlaid with cortical depth bins approximating cortical layers (superficial, middle, and deep) from an equi-volume model (see Methods). The cortex is mapped within the region of V1 with voxel eccentricity values 0–3°. **B** Classifier decisions in V1 over the time interval measured at 8 and 10 seconds after the rotation onset for the presented, mentally rotated and unused gratings in the outer cortical bins (average of the superficial and deep bin) and the middle cortical bin ($N = 23$ participants; see Supplementary Fig. 3 for detailed analysis within an extended time interval and Supplementary Fig. 4 for analyses across all time points). Perceptual contents were more strongly represented at the middle cortical depth, whereas mentally rotated contents were dominant at the outer cortical bin. **C** Comparing classifier decisions between all three cortical bins (superficial vs. middle vs. deep) reveals that the difference between perceived and rotated contents is most pronounced between the middle and superficial depths. All error bars denote standard error of mean over subjects. $+p < 0.09$, $*p < 0.05$, $**p < 0.01$, $***p < 0.001$.

was pre-selected based on previous studies[4,5] where the mentally rotated gratings could be decoded starting from 8 seconds following the rotation instruction (also see Methods for further clarification). Previous fMRI studies[14,20], however, utilized different time intervals and experimental tasks to show the distribution of feedforward and feedback signals in cortical depth (for the perception signal decoding based on orientation localizer task in our study see Supplementary Fig. 2). Building on this previous work, our main goal here was to disentangle concurrent representations of perceived and mentally rotated contents across cortical depth, even when they are represented in a spatially and temporally overlapping way.

First, we hypothesized that representations of mentally rotated and perceived contents should emerge at the outer and middle cortical depth bins, respectively. To test this hypothesis, we compared the mental rotation and perception signals to the unused grating in the average of the outer bins and in the middle bin (Fig. 2B, see Methods for the clarification for the choice of the baseline). We found significantly more classifier choices for the rotated grating in the outer cortical bins ($t_{22} = 4.6$, $p = 0.0001$, Cohen's d = 0.97, FDR-corrected for the number of cortical bins), but not at the middle depth ($t_{22} = 0.2$, $p = 0.9$). Classifier choices for the presented grating were significantly more frequent than for the unused grating at the middle depth ($t_{22} = 2.8$, $p = 0.014$, Cohen's d = 0.59) as well as in the outer depth bins ($t_{22} = 2.7$, $p = 0.006$, Cohen's d = 0.56). In sum, mentally rotated contents reached significance only in the outer cortical bins, whereas perception signal was present at all the cortical bins.

To compare mental rotation and perception signals across cortical depth bins, we ran a repeated-measures ANOVA with factors Signal Type (perception vs. mental rotation) and Cortical Depth (middle vs. outer cortical bins) (Fig. 2B). The analysis revealed a significant interaction ($F_{1,22} = 10.95$, $p = 0.0045$). More information about the rotated orientation was found in the outer cortical bins than at the middle depth ($t_{22} = 2.52$, $p = 0.0096$, Cohen's d = 0.53). In contrast, more information about the perceived orientation was present at the middle bin than in the outer

bins ($t_{22} = 2.8$, $p = 0.0052$, Cohen's d = 0.58). Similar distribution of feedforward signal was observed in V2 (Supplementary Fig. 5). We conclude that information about perceived and mentally rotated contents in V1 is spatially separated across cortical depth bins and functionally corresponds to cortical layers: the outer bins in our study mainly represented mentally generated contents and the middle bin selectively encoded sensory information.

As superficial and deep cortical depth bins were aggregated in the aforementioned analyses, we performed an additional analysis comparing all three depth compartments (deep vs. middle vs. superficial). A repeated-measures ANOVA again revealed a significant interaction between Signal Type and Cortical Depth ($F_{2,44} = 5.3$, $p = 0.0085$) (Fig. 2C; for the univariate analysis results see Supplementary Fig. 6). Unpacking this, mentally rotated orientation was more strongly represented in the superficial cortical bin than in the middle one ($t_{22} = 2.7$, $p = 0.02$, Cohen's d = 0.56) and more strongly in the deep bin than at the middle cortical depth (again at the trend level; $t_{22} = 2.7$, $p = 0.08$, Cohen's d = 0.35). In contrast, the perceived orientation was more strongly represented in the middle bin than at the superficial ($t_{22} = 2.9$, $p = 0.014$, Cohen's d = 0.6) and deep ones (at the trend level; $t_{22} = 1.8$, $p = 0.06$, Cohen's d = 0.38). No statistically significant difference between the deep and superficial layers was found when analyzing perception ($t_{22} = 0.86$, $p = 0.19$) or mental rotation signals ($t_{22} = 0.6$, $p = 0.27$); however, we acknowledge that these and other non-significant differences in our study might have reached significance with larger samples. Mentally rotated contents were thus represented in both outer depth bins, albeit weaker dissociations emerged between the middle and the deep cortical bins.

## Discussion

Harnessing the fine-grained resolution of 7 T fMRI, we were able to resolve the functional segregation of signals underlying mentally rotated and perceived contents in V1: perceptual signals were dominant at the middle depth of V1, whereas mentally rotated contents were found in the superficial and deep bins.

While our results are consistent with the previous fMRI studies at the standard resolution showing that V1 houses representations of both perceived and mentally rotated contents[4,5] (Supplementary Fig. 1B), our findings provide the first functional explanation for externally induced and internally generated representations during mental rotation overlapping on a 2D map of cortex, yet functionally distinct in a 3D cortical model.

The functional separation of the contents of perception and mental rotation into cortical depth bins follows the neuroanatomy of V1, which is characterized by bottom-up connections terminating in the middle cortical layer and top-down connections terminating in the outer layers[6–11]. Our results demonstrate that this anatomical differentiation between feedforward and feedback connections directly maps onto activity time courses during a cognitive task. A similar mapping between fine-scale cortical architecture and bottom-up and top-down information flow underpins basic visual functions such as illusory perception[20,21] and visual expectations[22]. Although different involvements of superficial and deep cortical depths were reported in these studies, they consistently highlight a laminar separation between the middle and the outer cortical subdivisions in V1. Specifically, a more pronounced representation of the feedback signal at the superficial cortical depth in our study could potentially result from different underlying processes. First of all, this finding is consistent with the previous studies[15,20] where feedback signal was measured in the presence of physical stimuli. In our experiment, perceptual stimuli were only briefly shown at the trial onset, but perception contents were reliably represented in the brain activity patterns throughout the trial duration and thereby could impact the depth distribution of the feedback signal. Another possibility is that fMRI measurements obtained with gradient-echo sequence in our study could be biased towards superficial cortical depth due to close proximity to pial veins (effect of draining veins)[23–25] resulting in comparatively stronger dissociations between the middle and superficial cortical bins.

We demonstrated that perception contents are more strongly represented at the middle depth when estimated concurrently with mentally rotated contents, while no such difference was observed in the absence of feedback manipulation (Supplementary Fig. 2). This result points to a separation of feedforward and feedback signals by cortical depth in V1 possibly in order to avoid confusion between the two information streams. A similar result was obtained in a recent study, which concurrently manipulated feature-based attention (top-down) and stimulus contrast (bottom-up)[15], suggesting that the observed functional signal-by-layer separation may be a general cortical mechanism.

The involvement of V1 in dynamically representing internally generated contents invites a redefinition of the region's role for visual cognition. Our results support the view that V1 is not only a sophisticated feature processor for sensory input passing through, but rather a high-resolution buffer that can be dynamically updated through connections from higher-order areas. The view of V1 as a dynamic "blackboard"[26,27] is consistent with studies reporting V1 activations when stimulation is absent[28–30], when participants direct attention without visual inputs[31], and after stimulating other modalities[32], even in the blind[33,34].

Although supporting the dynamic "blackboard" view, our results confront the idea of a 'perception-like' nature of mental images[27,35], indicating that feedforward and feedback mechanisms manifest in distinct neural populations of V1. This dissociation of information flow by cortical depth questions the idea of 'shared representation' for mental rotation and perception pointing towards a necessity for further clarification of the properties that are common or instead uniquely owned by each

process. For this, future studies are needed to systematically compare perceived and mentally rotated representations in the middle[36] and outer cortical layers, for example, through contrasting perception- and imagery-induced retinotopic maps of low-level features such as horizontal/vertical meridian, foveal/parafoveal cortical divisions, or orientation discrimination biases.

Beyond the spatial separation of representations of perceived and mentally rotated contents, the laminar organization of feedforward and feedback information may also facilitate interactions between these signals. Processing bottom-up and top-down signals in close physical proximity on the cortical surface optimizes crosstalk that is essential for a large set of cognitive functions such as figure-ground segregation[16,37], surround suppression[38], visual attention[17–19], and visual short-term memory[12,14,39]. In fact, predictive coding accounts suggest that a multitude of brain processes depend on such interactions[40–42], rendering the laminar separation of feedforward and feedback information a candidate for an implementation strategy for various other brain functions.

Layer-specific fMRI is an emerging technique requiring further procedure stabilization and refinement of analysis to ensure that obtained results are not impacted by motion artifacts (see Methods), draining veins effect or data acquisition methods potentially introducing resolution losses[43]. We acknowledge a potential influence of these factors on our results, and future studies using alternative protocols[44] to alleviate potential confounds are required to accrue additional evidence.

Although our study highlights that V1 represents internally generated contents, a firm link between such V1 representations and the subjective quality of mental images is yet to be established. Future studies could directly link fine-grained cortical feedback patterns to an individual's ability to successfully conjure up mental representations during mental rotation[45–47] and mental imagery[48,49]. Such studies may also help to reveal imbalances between feedforward and feedback signals that lead to aphantasia[50–52], hallucinations or other perceptual disturbances[53,54]—and eventually treat these symptoms in the future.

Together, our results elucidate how the contents of perception and mental rotation are simultaneously represented in different cortical compartments of V1. Our findings thereby highlight that early visual cortex is not only involved in the analysis of sensory inputs but is also recruited during dynamic visual cognition. Separating these different functions across cortical depth may reflect a general strategy for implementing multiple cognitive functions within a single brain region.

## Methods

**Participants.** Twenty-five healthy adults (age Mean ± SD: 29 ± 5.7; 9 female) participated in the study. All participants had normal or corrected-to-normal vision. Participants gave their written informed consent for participation in the study as well as for publicly sharing all obtained data in pseudonymized form. They received monetary reimbursement for their participation. The study was approved by the ethics committee of the Otto-von-Guericke-University Magdeburg, Germany. Two participants had to be excluded due to aborted data collection and artifacts in the anatomical T1-weighted image, respectively. All analyses were conducted on the remaining 23 participants.

We chose the number of subjects that was similar or exceeded the sample sizes of previous 7 T studies investigating feedback signals with laminar separation[14,15,18,20,38]. With our sample size ($N = 23$) and at statistical power 80%, a medium size effect is detectable in our study ($d = 0.62$, paired-samples two-sided $t$-test).

**Stimuli.** Stimuli were grayscale luminance-defined sinusoidal gratings generated using MATLAB (MathWorks, Natick, MA) in conjunction with the Psychophysics Toolbox[55]. The gratings were presented in an annulus (outer diameter: 6.7° of visual angle, inner diameter: 1.3° of visual angle) surrounding a central fixation

point. The gratings had a spatial frequency of 2 cpd (12.34 Hz) and a Michelson contrast of 100%. Stimuli were displayed on an LCD projector (DLR-RS49E, JVC Ltd.) on a rear-projection screen positioned behind the head coil within the magnet bore. Participants viewed the screen through a mirror attached to the head coil.

### Experimental procedure

*Training procedure.* Before entering the MRI scanner, participants underwent a training procedure, which comprised minimum 4 runs for all the participants, with 6 being the maximum number in case participants needed more time to learn how to perform the task. At the start of each trial, participants briefly saw a randomly oriented grating (Fig. 1A). The stimulus presentation was followed by a mask comprising the intersection of three gratings (15°, 75°, and 135°) at random phase. A subsequently presented task cue indicated which direction participants had to rotate the presented stimulus grating in their mind's eye. Mental rotation could go either clockwise or counterclockwise (as indicated by arrow direction), and for 60° or 120° (as indicated by the number of arrows). After a 6 second rotation period a probe grating was shown. The probe comprised the grating shown in the beginning of the trial, rotated in accordance with the cue instruction. Additionally, the grating was slightly tilted clockwise or counterclockwise; the amount of additional tilt was regulated using staircase procedure (described below). The participants' task was to indicate the direction of difference between the probe grating and the mentally rotated grating. After each trial, participants received a 1 second feedback about their performance. The inter-trial interval was 2 seconds. Each training run consisted of 36 trials and took 7 min 52 seconds. At the end of each run, participants received feedback about their average accuracy.

*Staircase procedure.* To maintain a sensitive accuracy range, the extent of additional tilt in the test stimulus, (compared to the orientation resulting from the mental rotation) was adjusted using a staircase procedure. The initial difference between the orientation resulting from mental rotation and probe grating was set at 20°. For each correct response in a given trial, the difference between the probe and the rotated grating was reduced by 0.5°, making orientation discrimination harder. Conversely, the difference was increased by 2° for each incorrect response, making discrimination easier. We imposed an upper limit of 40° on the orientation difference. The staircase procedure continued across the whole experiment, including the training runs and the fMRI experiment.

*Experimental task.* In the scanner, participants first underwent an anatomical scanning procedure, during which we acquired two T1-weighted anatomical scans, two PD-weighted contrasts and a T2-weighted contrast (described in more detail in 'Parameters of Data Acquisition'). The anatomical scanning procedure took ~40 min. During anatomical scan procedure participants were encouraged to rest and move as least as possible to reduce motion artefacts.

After that, participants continued to perform the task, which they were trained in beforehand (Fig. 1A), but with two major changes. First, participants did not receive feedback on their performance to increase the number of trials during scanning time. Second, the sample gratings shown at the beginning of each trial were no longer randomly oriented, but limited to 15°, 75°, or 135° orientation from the vertical axis. We limited the number of possible stimuli compared to the training session to increase signal-to-noise ratio per each sample grating and to enable signal differentiation at the level of cortical depth bins. In future studies, the use of richer stimulus sets may provide insight to whether the same neural processes, which govern the mental rotation tasks, are also performed on everyday objects during our daily lives.

We generated a cyclical design, that is, each of these orientations could be turned into one of the two other orientations on a circle defined by stimulus orientation (Fig. 1B). In effect, the presented and rotated gratings were different from each other on every trial, allowing us to independently assess encoding of perceived and mentally rotated contents.

Overall, there were three possible starting orientations (15°, 75°, or 135°), two directions of rotation (clockwise or counterclockwise) and two rotation magnitudes (60° or 120°), resulting in 12 unique trial types. The 12 unique trials were repeated three times within each run, resulting in a total of 36 trials. Trial order was fully randomized. In a nutshell, the experiment consisted of 6 runs, which each lasted 7 min 16 seconds.

We cannot ultimately exclude the possibility that participants realized how many stimuli were shown overall and only retrieved the relevant orientation from memory rather than performing the rotation task properly. Such strategies are a typical problem in mental rotation studies using a fixed number of repeating stimuli. However, our behavioural data provides direct evidence against this scenario: the response time data cannot be accounted for by the retrieval of fixed orientations (or orientation) labels from memory. We would like to add that after the experiment, we asked each participant how many orientations they had to rotate and none of them reported the real number of orientations in the stimulus set. Therefore, we believe that our participants were genuinely performing the mental rotation task.

*Orientation localizer task.* To select voxels most responsive to each of the three orientations shown in the experiment, participants finally completed an additional orientation localizer run. During this run, gratings with the three orientations (15°, 75°, and 135°) were shown in a block design in a pseudo-randomized order. In each block, one of the grating orientations was shown for 12 seconds, flickering at 4 Hz. Every three blocks (i.e. one repeat of the three orientations) were followed by a fixation block, which lasted 15 seconds. Participants had to monitor the fixation cross for occasional brief changes in color, to which they had to respond with a button press. Overall, we recorded data for 60 blocks (45 orientation blocks and 15 fixation blocks). The fixation dot changed 9–10 times per block at random time points, leading to ~144 changes, to which participants responded on average 94 ± 4% (Mean ± SD) of the time. The orientation localizer task was performed last in the experiment to ensure that participants did not notice that the orientations shown to them during prolonged periods in the localizer task (12 s) are the same three orientations as the ones briefly presented in the beginning of each trial of the main experiment. The localizer task took 12 min 49 seconds. The average time for completing the whole experiment was 115 min including anatomical scans.

### Parameters of data acquisition

MRI data were acquired using a 7 T Siemens whole-body MR scanner (7 T Classic, Siemens Healthineers, Erlangen, Germany) using a 32-channel receive head coil (Nova Medical Head Coil, Wilmington, MA, USA). Functional data were acquired with a T2*-weighted 2D gradient-echo EPI sequence (TR 2000 ms, TE 22 ms, 0.8 mm isotropic voxels, number of slices 30, 90° flip angle, $128 \times 128$ mm$^2$ FOV, GRAPPA acceleration factor 4, slice partial Fourier 5/8, coronal orientation, R » L phase encoding direction). Shimming was performed using the standard Siemens procedure. For the first half of the sample, anatomical data were acquired using a MPRAGE sequence with 0.8 mm isotropic resolution (TR 2500 ms, TE 3.05 ms, TI 1050 ms, flip angle 5, bandwidth 130 Hz/Px, $205 \times 205$ mm FOV, no GRAPPA applied, slice partial Fourier 6/8, base resolution 256, sagittal orientation, A » P phase encoding direction, scan time 9 min 20 sec). For the second half of the sample, an additional T1-weighted image[56] was acquired with the resolution of 0.7 mm isotropic voxels to provide a more precise delineation of cerebro-spinal fluid and grey matter at the segmentation stage (TR 2500 ms, TE 2.55 ms, TI 1050 ms, flip angle 5, bandwidth 320 Hz/Px, $224 \times 224$ mm FOV, GRAPPA factor 2, no partial Fourier, sagittal orientation, A » P phase encoding direction, time of acquisition 7 min 18 sec). Additional anatomical scans of ~18 min duration were acquired but not used here. During the functional data acquisition, geometric distortions were corrected using EPI-PSF-based distortion correction[57]. To correct for rigid-body motion, we applied prospective motion correction during the acquisition of both structural and functional scans[58]. It is an optical in-bore tracking system consisting of a single camera and a marker. In order to establish a rigid connection between the marker and the head, a custom-made dental mouthpiece of the six central teeth of the upper jaw has been crafted by the department of oral and maxillofacial surgery of the university hospital of the Otto-von-Guericke university, Magdeburg, Germany. The mouthpiece is equipped with an extension at which the marker is attached. Therefore, line of sight between the marker and the camera is never lost.

### Functional and anatomical data preprocessing

*Bias field correction and segmentation of the anatomical image.* The DICOM data were converted to NIfTI format using SPM 12 (Wellcome Trust Center for Neuroimaging, University College London). The volumes were bias field-corrected using an SPM-based customized script[56]. To implement cortical depth-specific analysis, we extracted grey matter segmentation for each subject. To do this, first we used the SPM 12 segmentation algorithm and then the brainmask was generated by adding up the white matter, grey matter and cerebro-spinal fluid masks. Then we applied the FreeSurfer (version 6.0.0) recon algorithm to perform segmentation of white matter, grey matter, generating their surfaces and a binary brainmask of the cortical ribbon (1 if the voxel falls into the ribbon, 0 otherwise (steps 5–31 of recon-all algorithm)). We ran the recon algorithm on the extracted brainmask from a T1-weighted image with a '-hires' flag for the data with resolution higher than 1 mm[56,59] After running the recon algorithm, the Freesurfer-generated grey and white matter segmentations were visually inspected in each participant, the borders between CSF and grey matter as well as grey matter and white matter were manually corrected within the region corresponding to the field of view of functional scans. To improve segmentation quality, we performed the Freesurfer segmentation algorithm not only on the T1-weighted image but also the T1-weighted image divided by the PD-weighted contrast[60]. However, the T1-weighted image after the division did not show essential advantages over using the data-driven bias field-corrected T1-weighted image. Therefore, for the further cortical depth separation we used the T1-weighted image without division.

*Cortical depth and ROI definition.* The grey matter segmentation acquired with Freesurfer was further utilized to obtain cortical depth-specific compartments. Deep, middle and superficial compartments were constructed using an equivolumetric model[61,62]. In order to analyze depth-specific activity in early visual areas, we applied a probabilistic surface-based anatomical atlas[63] to reconstruct the surfaces of areas V1, V2, and V3 separately for each region and subject. This is an atlas of the visual field representation (eccentricity and polar angle), and eccentricity values were used to select the foveal sub-part of the surface (0–3°). The extracted surface ROIs (V1–V3) were then projected into the volume space and intersected with the predefined cortical compartments. In this way, we obtained the

V1, V2, and V3 ROIs in the Freesurfer anatomical space at three predefined cortical depths.

*Image alignment check.* Functional volumes did not undergo any additional pre-processing. We did not perform realignment due to utilization of prospective motion correction. However, we ensured that the functional runs were well aligned with each other in each participant, which is required for multivariate pattern analyses of high-resolution fMRI data, by computing inter-run spatial cross-correlations of the signal intensities of the functional volumes. For two participants an intensity-based image registration algorithm in MATLAB was used to improve inter-run alignment until the inter-run correlations were at least $r > .9$ on average. The resulting average spatial correlation of experimental runs was (Mean ± SD) $0.986 ± 0.01$, with the following motion parameters in the translation and rotation directions (Mean ± SD): x: $−0.1 ± 0.3$, y: $0.2 ± 0.3$, z: $−0.04 ± 0.3$, pitch: $−0.002 ± 0.003$, roll: $−0.002 ± 0.005$, yaw: $−0.000 ± 0.003$. Further, functional-anatomical alignments were checked visually to ensure that the functional scans were well aligned to the anatomical image at the location of the ROIs.

*Registration.* We linearly coregistered the extracted ROIs with predefined cortical depth compartments to the EPI volumes within each subject using the Symmetric Normalization (SyN) algorithm of ANTs[64]. Specifically, first, the T1-weighted anatomical image was registered using linear interpolation to the EPI volume averaged over all the functional runs. Next, we registered the ROIs with the predefined cortical depths to the EPI volume using nearest neighbor interpolation and applying the coordinate mapping (with the voxel size resampled to the functional runs (0.8 isotropic)) obtained in the previous step (Fig. 2C). In the resulting ROIs the number of voxels per cortical depth (Mean ± SD) was the following in V1: $M_{deep} = 1158 ± 258$; $M_{mid} = 1073 ± 241$; $M_{super} = 936 ± 220$; in V2: $M_{deep} = 1096 ± 321$; $M_{mid} = 1049 ± 260$; $M_{super} = 901 ± 319$; in V3: $M_{deep} = 1123 ± 364$; $M_{mid} = 1019 ± 256$; $M_{super} = 855 ± 308$).

## Multivariate pattern analysis

*Data extraction.* Multivariate pattern analysis (MVPA) was performed in each subject individually. To prepare the EPI data for the MVPA, we first extracted activity patterns for each ROI with the predefined cortical depths from the functional images in the main experiment and orientation localizer run. Specifically, in each experimental run, we extracted voxel-wise activation values for three oriented grating conditions (15°, 75°, or 135°) and 12 trials for each condition across 5 TRs ($=10$ seconds), starting at trial onset. The EPI data from the orientation localizer run were also aggregated for the three oriented grating conditions (15°, 75°, or 135°) and 15 trials per each condition across 5 TRs of trial duration.

*Classification.* Multivariate pattern analysis (MVPA) was carried out using the linear support vector machine[65] (SVMs; libsvm: http://www.csie.ntu.edu. tw/~cjlin/libsvm/) with a fixed cost parameter ($c = 1$). We performed classification at each cortical depth and ROI independently in the following way. We trained the SVM classifier on the fMRI data from the orientation localizer run to discriminate between the three oriented gratings in each TR separately using all the trials (15 data points per orientation per training set). Next, we tested the SVM classifier using the EPI data from the main experiment (on each trial separately). Each TR in orientation localizer task used for the classifier training corresponded to the TR in the experimental trial used for the classifier testing. As a result, we extracted predicted labels (15°, 75°, and 135°) for every TR of all the trials in the main experiment (chance level 33.3%). Then, we compared the labels predicted by the SVM classifier with the oriented gratings actually presented, rotated or not used in each trial (as illustrated in Fig. 1C). The proportion of matches between the predicted grating label and the role of that grating in the trial was accumulated over trials for each of these three experimental conditions (presented, rotated, and unused gratings) to estimate their representational strength within each subject. Finally, the resulting estimates in a form of 23 (subjects) × 3 (presented, rotated, not shown grating) × 5 (TRs) matrix calculated for each ROI and cortical depth was subjected to statistical testing.

Note that in our paradigm the three orientations on each trial are not independent. The more information about one of the orientations is found (e.g. the perceived orientation), the less information is found about the other orientations (e.g. the rotated orientation). We therefore cannot compare classifier choices to "chance" level (i.e. 33%). Instead, we compare classifier choices for these orientations to the third, unused orientation. This procedure allowed us to estimate information about the perceived and rotated orientations independently from each other. For instance, if the representation of the perceived orientation is so strong that the classifier very often picks the perceived orientation, it may be that the classifier picks the rotated orientation in fewer than 33% of trials. However, this does not mean that there is no information about the rotated orientation: If there are still more classifier choices for the rotated orientation than the unused orientation, the rotated orientation is represented in the signal.

For the in-depth assessment of mental rotation contents, a critical time interval was chosen based on the previous studies[4,5] where mentally rotated representations could be decoded in the period 8–12 seconds after the rotation cue. In our study, we included time interval 8–10 s after the rotation cue since the measurement at 12 s was likely to carry the representation of a probe grating (shown at 8 s), while

the measurement at 10 s is too close to the presentation of the probe grating to be contaminated by it.

*Statistics.* We used repeated-measures analyses of variance (ANOVA) to test the main effect of Signal type (presented vs mentally rotated grating) in the trial and to test the interaction of Signal type and Cortical depths (deep and superficial vs. middle) (custom function rmanova2 derived by A. Schurger (2005) from Keppel & Wickens, "Design and Analysis", ch.18: https://de.mathworks.com/matlabcentral/fileexchange/6874-two-way-repeated-measures-anova). In cases where the assumption of sphericity was violated the $p$ values were corrected using a Huynh-Feldt correction (provided as an output of the same function). Significant interactions were followed up with paired-samples one-sided $t$-tests ($t$-test function in MATLAB) to analyze the effects in the assumed directions based on neuroanatomy and animal findings. To control for multiple comparisons across $t$-tests, we used FDR-corrections that assume independent or positively correlated tests[66]: these corrections allow for maintaining a low false-positive rate while providing reasonable power to find truly significant results.

## Data availability
The MRI and behavioural data that were used in this study are available: https://osf.io/3x9fk/?view_only=dd7d8e9462694501a60a4dd308fd9354.

## Code availability
MATLAB source code for LIBSVM toolbox is available online (libsvm: http://www.csie.ntu.edu. tw/~cjlin/libsvm/). The code for data sorting, utilizing LIBSVM toolbox in the present study, and plotting the main result is available: https://github.com/IamPolina/7T_Mental_Rotation.git.

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

## Acknowledgements

P.I. is supported by the Berlin School of Mind and Brain PhD scholarship. D.K. and R.M.C. are supported by Deutsche Forschungsgemeinschaft (DFG) grants (KA4683/2-1, CI241/1-1, CI241/3-1). R.M.C. is supported by a European Research Council Starting Grant (ERC-2018-StG). N.W. is supported by the European Research Council under the European Union's Seventh Framework Programme (FP7/2007-2013)/ERC grant agreement n° 616905; the European Union's Horizon 2020 research and innovation programme under the grant agreement No 681094; the BMBF (01EW1711A & B) in the framework of ERA-NET NEURON. We thank the staff of the Department of Oral and Maxillofacial Surgery, University Hospital Magdeburg A.ö.R., Christian Zahl, Indra Griesau, and Christine Rohloff for creating individually custom-made removable dental braces. Computing resources were provided by the high-performance computing facilities at ZEDAT, Freie Universität Berlin.

## Author contributions

P.I., D.K. and R.M.C. designed the study, R.M.C., E.D., O.S. and N.W. supervised the study, P.I., R.Y., A.S., H.M. and F.L. acquired data. P.I., D.K., D.H. and F.L. analyzed the data. P.I., D.K. and R.M.C. wrote the original draft of the manuscript, R.Y., A.S., H.M., F.L., E.D., O.S. and N.W. reviewed and edited the manuscript.

## Funding

## Competing interests

The authors declare no competing interests.
