## [Peer Review File · Communications Biology]

Reviewers' comments:

Reviewer #1 (Remarks to the Author):

Perceived and mentally rotated contents are differentially represented in cortical layers of V1 - Review

The study by Imashchinina et al. aims to find anatomical separation of distinct functional signals related to visual perception (bottom-up) and mental rotation (top-down) in early visual areas across cortical depths. The authors acquired high-resolution fMRI scans while participants performed a mental rotation task. The task consisted of mentally rotating a grating by a fixed amount in a specific direction. Using MVPA analysis they could show that visual perception signal is represented in the middle layers of V1. On the other hand, the outer layers, i.e., superficial and deep layers, represent the mentally rotated signals. According to the authors, this laminar separation of perceived and mentally rotated contents in V1 is the reason why these two functionally different types of content in one cortical area of the brain are not confused with each other.

The study contributes to an already existing body of evidence about the dissociation between bottom-up and top-down signals across the cortical depth. This has already been shown for working memory (Lawrence et al., 2018) attention (Lawrence et al., 2019), illusion perception (Kok et al., 2016; Marquardt et al., 2020) and missing information (Muckli et al., 2015) in the visual domain. Here the authors demonstrate the effect of the mental rotation task. Among the strengths of the study is a relatively large sample of subjects (n=23), and an elegant experimental design (although it seems to be adapted from a previous study of Albers et al. (Albers et al., 2013)). My main concerns are about the data analysis approaches and data presentation, which I list in detail below.

Major comments

1. Analysis method

I don't understand the reasons for using the multivariate rather than the univariate analysis in this experiment. The advantage of high-resolution fMRI is not only in detecting layer-related, but also column-related signals. The main assumption of the study is based on the orientation-selectivity in V1, so why not identify voxels selective to each of the three orientations, and analyze their responses? Similar approach has been taken by (Lawrence et al., 2019) and Lawrence et al., (2018) at 7T or by (Pajani et al., 2015) even at 3T. The authors should either present univariate results instead of or in addition to multivariate, or present solid arguments for why such analysis is not plausible.

2. Time-resolved analysis

I don't understand the choice of the main analysis window 6-10 seconds after the trial onset. Authors motivate this time window by citing the previous studies (lines 118-119). However, looking at the cited studies, e.g. at Albers et al., (2013) there are two issues. First, the best time point for decoding the presented and mentally rotated content differs (presented: 4 s after trial onset, rotated: 8-16 s after trial onset). So one can't optimally represent both processes by using only one time point. Second, Albers et al analyzed the time course of the whole trial and showed that mentally rotated content can be decoded *before* the probe onset (10 s in Albers et al). In the current study, the time window of 6-10 seconds includes the presentation of the test grating (at 8 s), which is simply wrong. The orientation of the test grating is similar to that of the mentally rotated grating, so in principle the results can be explained by decoding the test grating orientation.

The Extended Figure 3 shows the decoding over time (collapsed over layers), which does not replicate the Albers et al effect. So it is not clear after all how the temporal window was chosen.

I would ask the authors to show the decoding analysis for each time point, for each depth, and for each condition, as they promise in lines 94-95 in the main text.

Related to this, I disagree with the statement in line 617 "these signals are indistinguishable at the spatial resolution of standard fMRI recordings", as it contradicts Albers et al. findings, who used 3T and 3 mm voxels, but could show the dissociations of the signals across time.

3. What is the reason for comparing the decoding of presented and mentally rotated grating with the unused grating rather than with chance, as was done in Albers et al. (2013)? The higher the proportion of presented and mentally rotated predictions, the lower the proportion of unused predictions, so this comparison is artificially inflated.

Minor comments:

4. The reported effects seem quite small compared to previous studies, that showed around 50% decoding accuracy for 33% chance level. The authors should report effect sizes for significant effects, which is a good scientific practice.
5. Some additional discussion about the superficial layer driving the significance of the outer layer should be in the discussion. What are the implications and how is it consistent with the previous findings? What about the venous draining and the possible confounding effects of it on the signal acquisition (Uludağ & Blinder, 2017)?
6. I would ask the authors to be more cautious about referring to "layers" in general, or to a specific "middle layer" throughout the text, since it remains unclear how fMRI signals at different depth correspond to cortical laminae.
7. 64: Figure 1C: numbering the x-axis might improve the readability
8. 86: "non-anatomically defined" – what does it mean?
9. 104: Figure 2: labels B and C are so high up that they are easily overlooked
10. 145: statistical test results: I think it should be $t=2.8$ instead of $t=1.8$
11. 192: "figure-ground segregation" (add "-")
12. 207: You can also cite a recent review by (Stephan et al., 2019)
13. 407: fMRI acquisition: please report slice orientation and phase encoding direction
14. 418: The authors used PSF-based distortion correction method. This method was recently shown to introduce excessive blurring (stronger than the fieldmap-based method) (Bause et al., 2020). The authors should address the potential negative effects (resolution loss) of this method on the data.
15. 434: "recon-all steps 5-31" - it is not quite clear what the authors are referring to here. It would be more helpful to list these steps and/or briefly explain what they do
16. 435: "specific options for the data with resolution higher than 1 mm" - which options exactly? If the authors mean the "- hires" flag, they should cite (Zaretskaya et al., 2018)
17. 436: Which SPM12 skull strip algorithm? Also, I don't understand why a skull stripping was need if the brain mask was already generated from the SPM segmentation (as stated in lines 431-433). Maybe the sentence in 435-437 is just redundant?
18. 445: do the authors mean "former" instead of "latter" here
19. 452: the surface-based atlas was probably applied to the reconstructed surfaces rather than to the image itself; please adjust the wording
20. 452: I think the readers would appreciate an explanation that this was an atlas of the visual field representation (eccentricity and polar angle), and that eccentricity values were used to select the foveal sub-part of the surface
21. 481: as far as I understood, the anatomically defined masks were in voxel space. How were the functional images sampled from the anatomical ones, given that in some participants, the resolution of the structural scans was different, and in the other participants with 0.8 mm anatomy the two voxel grids were probably not aligned
22. 481: did the authors forgot to mention, or was there really no preprocessing of fMRI data (not even a high-pass filtering?). If so, perhaps it should be explicitly stated, because it is quite unusual.
23. 508 onwards: which software was used for statistical tests? Why FDR correction for multiple comparisons was used? This is not a typical multiple comparison correction method in the context of an ANOVA
24. 588: "between subject SEM" – the authors mean the "standard error of the mean over subjects"?

References

- Albers, A.M., Kok, P., Toni, I., Dijkerman, H.C., de Lange, F.P., 2013. Shared Representations for Working Memory and Mental Imagery in Early Visual Cortex. *Curr. Biol.* 23, 1427–1431. <https://doi.org/10.1016/j.cub.2013.05.065>
- Bause, J., Polimeni, J.R., Stelzer, J., In, M.-H., Ehses, P., Kraemer-Fernandez, P., Aghaeifar, A., Lacosse, E., Pohmann, R., Scheffler, K., 2020. Impact of prospective motion correction, distortion correction methods and large vein bias on the spatial accuracy of cortical laminar fMRI at 9.4 Tesla. *Neuroimage* 208, 116434. <https://doi.org/10.1016/j.neuroimage.2019.116434>
- Kok, P., Bains, L.J., van Mourik, T., Norris, D.G., de Lange, F.P., 2016. Selective Activation of the Deep Layers of the Human Primary Visual Cortex by Top-Down Feedback. *Curr. Biol.* 26, 371–6. <https://doi.org/10.1016/j.cub.2015.12.038>

Lawrence, S.J., Norris, D.G., de Lange, F.P., 2019. Dissociable laminar profiles of concurrent bottom-up and top-down modulation in the human visual cortex. *Elife* 8. <https://doi.org/10.7554/eLife.44422>

Marquardt, I., De Weerd, P., Schneider, M., Gulban, O.F., Ivanov, D., Wang, Y., Uludağ, K., 2020. Feedback contribution to surface motion perception in the human early visual cortex. *Elife* 9. <https://doi.org/10.7554/eLife.50933>

Muckli, L., De Martino, F., Vizioli, L., Petro, L.S., Smith, F.W., Ugurbil, K., Goebel, R., Yacoub, E., 2015. Contextual Feedback to Superficial Layers of V1. *Curr. Biol.* 25, 2690–5. <https://doi.org/10.1016/j.cub.2015.08.057>

Pajani, A., Kok, P., Kouider, S., de Lange, F.P., 2015. Spontaneous Activity Patterns in Primary Visual Cortex Predispose to Visual Hallucinations. *J. Neurosci.* 35, 12947–12953. <https://doi.org/10.1523/JNEUROSCI.1520-15.2015>

Stephan, K.E., Petzschner, F.H., Kasper, L., Bayer, J., Wellstein, K.V., Stefanics, G., Pruessmann, K.P., Heinzle, J., 2019. Laminar fMRI and computational theories of brain function. *Neuroimage* 197, 699–706. <https://doi.org/10.1016/j.neuroimage.2017.11.001>

Zaretskaya, N., Fischl, B., Reuter, M., Renvall, V., Polimeni, J.R., 2018. Advantages of cortical surface reconstruction using submillimeter 7 T MEMPRAGE. *Neuroimage* 165, 11–26. <https://doi.org/10.1016/j.neuroimage.2017.09.060>

Reviewer #2 (Remarks to the Author):

This is a very interesting study that makes a good contribution. I just have a few questions/concerns.

Focus on Mental Rotation?

The title of the paper suggests a focus on mentally rotated representations. This may lead the reader to expect a study looking at the representations during the rotation. However, it seems more likely that the current study is looking at the representation after the rotation or of a retrieved representation (imagined or internally generated). Therefore, I think it might be better to remove the focus on rotation from the title and elsewhere in the paper. Given that only a few orientations were ever presented, it seems possible that participants learned the responses and did not require mental rotation on each trial. I see that the RT data supports a mental rotation interpretation, but I don't find it entirely convincing. In any case, I don't think the assumption of mental rotation is needed to support the paper's conclusions, if the conclusion is that imagined, or internally generated representations are maintained in different layers of V1 than perceived representations. Maybe I am missing a critical part of the argument, but it seems to me that the same thing would be predicted for a task that required participants to recall a given item from memory and maintain it for a later test as the current "mental rotation task". If so, and the V1 representations are not specific to any sort of active mental rotation, then easing away from the focus on mental rotation may be best.

Related to this is the timing of the classifier analysis. The authors note that "To test this hypothesis, we compared perception and mental rotation signals between the average of the outer layers and the middle layer, from 6-10 seconds after trial onset (i.e., 2 TRs), as done in previous studies". It would be helpful to have this timing indicated on Figure 1A. My interpretation is that the analysis begins 4 seconds into the "mental rotation interval" and then covers the time when the test grating is presented. First, does the end of the "mental rotation interval" encompass mental rotation or the maintenance of the rotated or retrieved representation? Second, is it problematic to include the test grating presentation time of the trial given that during this time a new stimulus is being perceived? Including a sentence or two about why this timing was chosen other than stating that this what was done in previous research may be helpful to the reader that is unfamiliar with the previous research.

Power

Why did the authors choose to collect data from 23 participants? Was a power analysis conducted?

Localizer concerns?

It is explained that the localizer task consisted of presenting the gratings for 12 seconds and then also blocks of fixation for 15 seconds. What is the function of the fixation block? When is the data from localization task collected, only during the blocks when the gratings were presented or also during the fixation blocks? I am wondering if it matters if the localization task uses perception activation rather than internally-generated activation.

Are there any concern about doing the localizer last, after the experimental runs? Could the experimental runs change how the orientations are perceived? Might there be more top down influence on perception based on the experience of encoding, rotating and retrieving the orientations in the experimental portion of the experiment. Would the authors predict anything different if the localizer was completed before the experimental task?

Length of Experiment

Are there any concerns about fatigue in the experiment? The authors report that "The average time for completing the whole experiment was 75 minutes excluding anatomical scans." The anatomical scan took 40 minutes. So, the whole experiment took 115 minutes, correct? Why not report the whole time instead of reporting the time minus the anatomical scan?

What does the participant do during the anatomical scan?

Reviewer #3 (Remarks to the Author):

The Ms deals with an interesting issue and I think that it can contribute to better understand the role of V1 in perception and imagery processes.

Specifically, the study investigates if perceived and mentally transformed stimuli during a mental rotation processing can reflect a laminar-specific activation in V1 in response to visuo-spatial information. The findings report high activations in the middle layer of V1 for perceived stimuli, and activations involving the superficial and deep layers of V1 for imagined rotations. Moreover, the study takes advantage of high-resolution fMRI (7T) and well-designed cognitive paradigm/tasks. However, in its present form it suffers of a series of problems, which need to be addressed.

1. I am not sure if the sample size is in line with other investigations. However, it would be more appropriate to have a more objective criterion (power analysis), explained on the basis of test conditions and variables.
2. A concern regards the choice of degrees of mental rotation: only 3 angular positions with a high level of chance level (as also recognized by the authors) and the low number of trials. This undermines the methodological robustness of the experimental task used. This point needs to be deeply addressed.
3. The use of more mental rotations would have been beneficial in the comparison with the long tradition of research on mental rotations (e.g. Zacks, 2008)
4. Given the limited number of angular degrees and trials, I wonder if the authors can exclude that the participant did not use other types of visuospatial or memory strategies to perform the task.
5. It would be interesting to check for lateralization of the mean signal of each hemisphere (e.g. Roberts et al 2003; Frings et al 2006 and others).
6. Please give more details on the False Discovery Rate method criterion.
7. It's not clear to me if in the present paradigm the fixation cross could be used as an external reference point for completing the perceived or even imagined rotations. The Authors could address this concern.

We thank all the reviewers for the valuable input. We provide a structured reply below. The reviewers' comments are italicized, our replies are written as a plain text and citations from the manuscript are highlighted in grey.

Reviewer #1 (Remarks to the Author)

The study by lamshchinina et al. aims to find anatomical separation of distinct functional signals related to visual perception (bottom-up) and mental rotation (top-down) in early visual areas across cortical depths. The authors acquired high-resolution fMRI scans while participants performed a mental rotation task. The task consisted of mentally rotating a grating by a fixed amount in a specific direction. Using MVPA analysis they could show that visual perception signal is represented in the middle layers of V1. On the other hand, the outer layers, i.e., superficial and deep layers, represent the mentally rotated signals. According to the authors, this laminar separation of perceived and mentally rotated contents in V1 is the reason why these two functionally different types of content in one cortical area of the brain are not confused with each other.

The study contributes a to an already existing body of evidence about the dissociation between bottom-up and top-down signals across the cortical depth. This has already been shown for working memory (Lawrence et al., 2018) attention (Lawrence et al., 2019), illusion perception (Kok et al., 2016; Marquardt et al., 2020) and missing information (Muckli et al., 2015) in the visual domain. Here the authors demonstrate the effect the mental rotation task. Among the strengths of the study is a relatively large sample of subjects (n=23), and an elegant experimental design (although it seems to be adapted from a previous study of Albers et al. (Albers et al., 2013). My main concerns are about the data analysis approaches and data presentation, which I list in detail below.

We thank reviewer 1 for the very valuable and helpful input. We believe that the comments helped to improve the manuscript considerably.

Major comments

1. Analysis method

I don't understand the reasons for using the multivariate rather than the univariate analysis in this experiment. The advantage of high-resolution fMRI is not only in detecting layer-related, but also column-related signals. The main assumption of the study is based on the orientation-selectivity in V1, so why not identify voxels selective to each of the three orientations, and analyze their responses? Similar approach has been taken by (Lawrence et al., 2019) and Lawrence et al., (2018) at 7T or by (Pajani et al., 2015) even at 3T. The authors should either present univariate results instead of or in addition to multivariate, or present solid arguments for why such analysis is not plausible.

We thank the reviewer for the suggestion. We agree that our analysis capitalizes on orientation-selectivity in V1, but we do not see this as a reason to strongly prefer univariate over multivariate analyses. We however ran the suggested univariate analysis on our data, using the same analysis time interval as in the multivariate analysis (see Fig.1). In the univariate analysis,

the signal X depth interaction was significant (3X2 interaction: $F(2,44)=3.34$, $p=0.04$; 2X2 interaction: $F(1,22)=6.5$, $p=0.02$), showing stronger rotation information in the superficial and deep cortical bins and stronger perception information at the middle depth. Overall, the results of the univariate and the multivariate approaches thus converge. We have added the univariate results to the Supplementary Information (reproduced below) and refer to it in the main MS (lines 738-752):

Supplementary Figure 6. Signal-by-Depth interaction analyzed with a univariate approach. We sorted the voxels in each cortical depth by their preference towards each of the three grating orientations based on the voxel activations in the orientation localizer task. We then picked the 300 most orientation-preferring voxels for each of the three orientations. To quantify the rotation signal, we then calculated the difference between the activation for rotated grating and two other gratings (non-preferred) within the voxels which preferred the rotated grating condition. To quantify the perception signal, we calculated the difference between the activation for perceived grating and two other gratings within the voxels which preferred the perceived grating condition. Prior to the analysis, the voxel responses both in the orientation localizer task and in the main experiment were high-pass filtered using spm12. We found a signal X depth interaction (3X2 interaction: $F(2,44)=3.34$, $p=0.04$; 2X2 interaction: $F(1,22)=6.5$, $p=0.02$), showing stronger rotation information in the superficial and deep cortical bins and stronger perception information at the middle depth. In detail, perception signal was significantly above 0 at the middle cortical depth ($t(22)=1.8$, $p=0.05$, Cohen's $d=0.37$) and trended to be stronger than in the outer bins (middle vs. deep bins: $t(22)=1.6$, $p=0.06$ Cohen's $d=0.33$; middle vs. superficial bins: $t(22)=1.6$, $p=0.06$, Cohen's $d=0.34$). The rotation signal was significantly above 0 in the deep bin ($t(22)=1.9$, $p=0.03$, Cohen's $d=0.4$) and stronger in the outer bins than the middle bin (middle vs. deep bin: $t(22)=2.6$, $p=0.007$, Cohen's $d=0.55$; middle vs. superficial bin: $t(22)=1.8$, $p=0.04$, Cohen's $d=0.37$).

2. Time-resolved analysis

I don't understand the choice of the main analysis window 6-10 seconds after the trial onset. Authors motivate this time window by citing the previous studies (lines 118-119). However, looking at the cited studies, e.g. at Albers et al., (2013) there are two issues. First, the best time point for decoding the presented and mentally rotated content differs (presented: 4 s after trial onset, rotated: 8-16 s after trial onset). So one can't optimally represent both processes by using only one time point. Second, Albers et al analyzed the time course of the whole trial and

*showed that mentally rotated content can be decoded *before* the probe onset (10 s in Albers et al). In the current study, the time window of 6-10 seconds includes the presentation of the test grating (at 8 s), which is simply wrong. The orientation of the test grating is similar to that of the mentally rotated grating, so in principle the results can be explained by decoding the test grating orientation.*

The Extended Figure 3 shows the decoding over time (collapsed over layers), which does not replicate the Albers et al effect. So it is not clear after all how the temporal window was chosen. I would ask the authors to show the decoding analysis for each time point, for each depth, and for each condition, as they promise in lines 94-95 in the main text.

Related to this, I disagree with the statement in line 617 “these signals are indistinguishable at the spatial resolution of standard fMRI recordings”, as it contradicts Albers et al. findings, who used 3T and 3 mm voxels, but could show the dissociations of the signals across time.

Before addressing these comments, we would like to clarify that we seem to name the same time points in different ways. Significant mental rotation signals in the study by Albers et al. (2013) were found in the 8-12 seconds interval, that is, the time window including measurements at 8, 10 and 12 seconds. In our study, we analyzed measurements at 8 and 10 s after the rotation cue onset; however, we referred to this interval as 6 to 10 seconds, as every measurement reflects the signal dynamics over the interval of 2 s TR (6-8 s and 8-10 s). To resolve this confusion, we adjusted the wording and the plots to represent specific time points exactly when the signal was acquired (2, 4, 6, 8 and 10 seconds after the trial onset, see Supp. Figure 1 in the revised manuscript).

In their comment, the reviewer raises four issues which we will address in the following:

- 1. Same time points in the trial cannot reflect the best decoding for perception and mental rotation.*

As the reviewer mentioned, we analyze both perception and mental rotation signals within the same time period in the main experiment. We chose this time period to prioritize the estimation of mentally rotated representation. We agree that this time period may not be optimal for estimating perception signals; however, perception signals are normally more strongly represented than feedback and can be decoded for an extensive duration. In this way, we aimed to disentangle spatially – as well as temporally – overlapping feedforward and feedback signals in cortical depth. We clarified this in the Results section of the manuscript (lines 103-108):

“Previous fMRI studies^{14,20} however utilized different time intervals and experimental tasks to show the distribution of feedforward and feedback signals in cortical depth (for the perception signal decoding based on orientation localizer task in our study see Supplementary Fig. 2). Building on this previous work, our main goal here was to disentangle concurrent representations of perceived and mentally rotated contents across cortical depth, even when they are represented in a spatially and temporally overlapping way.”

II. *Inclusion of the time point after the task grating presentation into the critical analysis time window is confounded with the perception signal from the task grating.*

We see that our analysis choice was not argued for clearly enough. Due to the sluggishness of the BOLD signal, a measurement taken two seconds after the onset of the probe grating most unlikely reflects the neural response to the probe grating. Our procedure is also equivalent to the

Albers et al. (2013) study using an analysis window that included BOLD signals shortly after the presentation of the task grating, too. We clarified and extended our explanation in the results and methods sections of the manuscript:

(lines 100-103; Results):

“We performed in-depth analyses in the time interval from 8 to 10 seconds (i.e., 2 TRs) after the rotation cue. This time interval was pre-selected based on previous studies⁴⁻⁵ where the mentally rotated gratings could be decoded starting from 8 seconds following the rotation instruction (also see Methods for further clarification).”

(lines 571-576; Methods):

“For the in-depth assessment of mental rotation contents, a critical time interval was chosen based on the previous studies (⁴⁻⁵ in the main references) where mentally rotated representations could be decoded in the period 8-12 seconds after the rotation cue. In our study, we included time interval 8-10 s after the rotation cue since the measurement at 12 s was likely to carry the representation of a probe grating (shown at 8 s), while the measurement at 10 s is too close to the presentation of the probe grating to be contaminated by it.”

III. *The Extended Figure 3 shows the decoding over time (collapsed over layers), which does not replicate the Albers et al effect. So it is not clear after all how the temporal window was chosen. I would ask the authors to show the decoding analysis for each time point, for each depth, and for each condition, as they promise in lines 94-95 in the main text.*

The results of our study do not fully replicate the study by Albers et al. (2013) but show a substantial overlap. The perception signal in both studies reached its peak at 4 seconds (see Figure 1). The mental rotation signal peaked between 6 and 10 seconds in our study and between 8 and 12 in the study by Albers et al. However, there is a difference in the temporal unfolding of perception signals: in our study the perceived orientation could be decoded 4 seconds into the trial, whereas in the study by Albers et al. (2013), the perception signal was not any more significant 4 seconds after the rotation cue.

We selected our analysis window in a principled a-priori way, before looking at the data. To provide a fuller characterization of the data, we now also report how perception and rotation signals unfold across time and across cortical depth (Supplementary Fig. 4, lines 718-723):

Supplementary Figure 4. Depth-specific time series of classifier decisions in area V1 plotted separately for feedforward (A) and feedback signals (B). As in the main analysis, classifier decisions for the perceived and rotated orientation were tested against decisions for the unused orientation (not shown here to avoid clutter). All error bars denote standard error of the mean over subjects. *: $p < 0.05$, **: $p < 0.01$, ***: $p < 0.001$ (uncorrected for multiple comparisons).

We added a reference to these data in the caption for the main results figure (Results section, lines 113-117):

“Classifier decisions in V1 over the time interval measured at 8 and 10 seconds after the rotation onset for the presented, mentally rotated and unused gratings in the outer cortical bins (average of the superficial and deep bin) and the middle cortical bin (see Supplementary Fig. 3 for detailed analysis within an extended time interval and Supplementary Fig. 4 for analyses across all time points).”

IV. *Related to this, I disagree with the statement in line 617 “these signals are indistinguishable at the spatial resolution of standard fMRI recordings”, as it*

contradicts Albers et al. findings, who used 3T and 3 mm voxels, but could show the dissociations of the signals across time.

We agree with the reviewer and we removed this sentence.

3. What is the reason for comparing the decoding of presented and mentally rotated grating with the unused grating rather than with chance, as was done in Albers et al. (2013)? The higher the proportion of presented and mentally rotated predictions, the lower the proportion of unused predictions, so this comparison is artificially inflated.

We thank the reviewer for this question, and we agree that a more thorough explanation of this choice is warranted.

Typically, experimental conditions are varied independently (e.g., 15 deg and 75 deg orientations can be used as perceptual input in the experiment, whereas 67 and 117 can be the results of mental rotation). In this case, successful decoding of the perception signal would have no effect on the decoding of the rotation signal, since rotation is manipulated through a different set of exemplars. Such experimental designs yield meaningful comparisons of each condition with interpretable chance level. However, as the reviewer noted, in our experimental design, a strong representation of the perception signal has an effect on how well the rotation signal can be decoded. As the same sets of exemplars are used in both conditions (15,75 and 135 deg), and the perceived and rotated orientations are never the same, a strong representation of the perceived orientation would in turn lead to fewer classifier decisions for the other two orientations (including the rotated orientation). This classifier behavior is an inherent and expected feature of our design.

Making inferences about a reliability of the signal presence in our design can be implemented via comparisons of the classifier choices for the perceived and rotated orientation to the third, unused orientation. If neither the perceived nor the rotated orientation are represented, each orientation should be picked equally often (and equally often as the unused grating's orientation). If there is only a representation of the perceived orientation, the classifier should most often pick the perceived orientation, and less (and equally) often pick the two other orientations. In this case, the perception signal should also be above the "chance" level of 33%, whereas the other two signals are not at "chance", but below (due to their dependencies).

The problem with comparing classifier choices to chance level arises when there is a representation of both the perceived and the rotated orientation. In this case, whether classifier choices for one of these two single conditions are above the 33% entirely depends on the relative strengths of the signals to each other. If perception and rotation signals are both present but one of them is stronger than the other, the estimation of the stronger signal happens at the expense of the weaker. Imagine a case where the perception signal strongly outweighs the rotation signal. Classifiers would almost always pick the perceived orientation, driving classifier picks for the imagined and the unused orientations strongly below 33%. Importantly, this does not mean that there can be no representation of the rotated orientation: If the rotated orientation is still chosen significantly more often than the unused orientation, that must be due to the rotated orientation being represented in the signal.

Comparing classifier choices to what seemingly is “chance” level is therefore not a viable option in our design.

We now mention the dependency among orientations as the reason for our analysis approach. We updated Methods section (lines 560-570):

“Note that in our paradigm the three orientations on each trial are not independent. The more information about one of the orientations is found (e.g., the perceived orientation), the less information is found about the other orientations (e.g., the rotated orientation). We therefore cannot compare classifier choices to “chance” level (i.e., 33%). Instead, we compare classifier choices for these orientations to the third, unused orientation. This procedure allowed us to estimate information about the perceived and rotated orientations independently from each other. For instance, if the representation of the perceived orientation is so strong that the classifier very often picks the perceived orientation, it may be that the classifier picks the rotated orientation in fewer than 33% of trials. However, this does not mean that there is no information about the rotated orientation: If there are still more classifier choices for the rotated orientation than the unused orientation, the rotated orientation is represented in the signal.”

Minor comments:

4. *The reported effects seem quite small compared to previous studies, that showed around 50% decoding accuracy for 33% chance level. The authors should report effect sizes for significant effects, which is a good scientific practice.*

This is a good suggestion. We have added Cohen’s d as a measure of effect size for all t-tests with significant results.

5. *Some additional discussion about the superficial layer driving the significance of the outer layer should be in the discussion. What are the implications and how is it consistent with the previous findings? What about the venous draining and the possible confounding effects of it on the signal acquisition (Uludağ & Blinder, 2017)?*

We added the following interpretations in the Discussion section (lines 175-184):

“Specifically, a more pronounced representation of the feedback signal at the superficial cortical depth in our study could potentially result from different underlying processes. First of all, this finding is consistent with the previous studies^{15, 20} where feedback signal was measured in the presence of physical stimuli. In our experiment, perceptual stimuli were only briefly shown at the trial onset, but perception contents were reliably represented in the brain activity patterns throughout the trial duration and thereby could impact the depth distribution of the feedback signal. Another possibility is that fMRI measurements obtained with gradient-echo sequence in our study could be biased towards superficial cortical depth due to close proximity to pial veins (effect of draining veins)²³⁻²⁵ resulting in comparatively stronger dissociations between the middle and superficial cortical bins.”

6. *I would ask the authors to be more cautious about referring to “layers” in general, or to a specific “middle layer” throughout the text, since it remains unclear how fMRI signals at different depth correspond to cortical laminae.*

We agree with the reviewer and we replaced “layers” with “cortical bins” throughout.

7. *64: Figure 1C: numbering the x-axis might improve the readability*

Thanks, done.

8. *86: “non-anatomically defined” – what does it mean?*

We removed this misleading phrasing and instead added a reference to the relevant Methods section.

9. *104: Figure 2: labels B and C are so high up that they are easily overlooked*

Thanks, corrected.

10. *145: statistical test results: I think it should be $t=2.8$ instead of $t=1.8$*

We checked and $t=1.8$.

11. *192: “figure-ground segregation” (add “-”)*

Thanks, done.

12. *207: You can also cite a recent review by (Stephan et al., 2019)*

Thanks, done.

13. *407: fMRI acquisition: please report slice orientation and phase encoding direction*

Thanks, done (lines 498, 503, 508).

14. *418: The authors used PSF-based distortion correction method. This method was recently shown to introduce excessive blurring (stronger than the fieldmap-based method) (Bause et al., 2020). The authors should address the potential negative effects (resolution loss) of this method on the data.*

This is an interesting point. In our study, we found a clear differentiation of responses across stimulus orientations and across cortical depths, which shows that the potential blurring introduced by PSF-based distortion correction was not fatal for the experiment. Nevertheless,

we agree that our choice of correction method requires discussion, so we added the reference to a paragraph in the Discussion section (Lines 218-223):

“Layer-specific fMRI is an emerging technique requiring further procedure stabilization and refinement of analysis to ensure that obtained results are not impacted by motion artifacts (see Methods), draining veins effect or data acquisition methods potentially introducing resolution losses⁴³. We acknowledge a potential influence of these factors on our results, and future studies using alternative protocols⁴⁴ to alleviate potential confounds are required to accrue additional evidence.”

15. 434: *“recon-all steps 5-31” - it is not quite clear what the authors are referring to here. It would be more helpful to list these steps and/or briefly explain what they do*

We added a more detailed description of the steps (Lines 486-490):

“Then we applied the FreeSurfer (version 6.0.0) recon algorithm to perform segmentation of white matter, grey matter, generating their surfaces and a binary brain mask of the cortical ribbon (1 if the voxel falls into the ribbon, 0 otherwise (steps 5-31 of recon-all algorithm)). We ran the recon algorithm on the extracted brainmask from a T1-weighted image with a ‘-hires’ flag for the data with resolution higher than 1 mm^{2,3}”

16. 435: *“specific options for the data with resolution higher than 1 mm” - which options exactly? If the authors mean the “-hires” flag, they should cite (Zaretskaya et al., 2018)*

Thanks, we added the citation.

17. 436: *Which SPM12 skull strip algorithm? Also, I don’t understand why a skull stripping was need if the brain mask was already generated from the SPM segmentation (as stated in lines 431-433). Maybe the sentence in 435-437 is just redundant?*

A brain mask was generated only once using the SPM12 segmentation procedure before submitting it to the Freesurfer algorithm. We agree that the sentence which the reviewer refers to is redundant, and we removed it.

18. 445: *do the authors mean “former” instead of “latter” here*

To improve readability, we restructured the sentence (lines 493-498):

“To improve segmentation quality, we performed the Freesurfer segmentation algorithm not only on the T1-weighted image but also the T1-weighted image divided by the PD-weighted contrast⁶. However, the T1-weighted image after the division did not show essential advantages over using the data-driven bias field-corrected T1-weighted image. Therefore, for the further cortical depth separation we used the T1-weighted image without division.”

19. 452: *the surface-based atlas was probably applied to the reconstructed surfaces rather than to the image itself; please adjust the wording*

20. 452: *I think the readers would appreciate an explanation that this was an atlas of the visual*

field representation (eccentricity and polar angle), and that eccentricity values were used to select the foveal sub-part of the surface

We added the following clarifications to the Methods section (lines 502-507):

“In order to analyze depth-specific activity in early visual areas, we applied a probabilistic surface-based anatomical atlas⁹ to reconstruct the surfaces of areas V1, V2 and V3 separately for each region and subject. This is an atlas of the visual field representation (eccentricity and polar angle), and eccentricity values were used to select the foveal sub-part of the surface (0-3°). The extracted surface ROIs (V1-V3) were then projected into the volume space and intersected with the predefined cortical compartments.”

21. 481: as far as I understood, the anatomically defined masks were in voxel space. How were the functional images sampled from the anatomical ones, given that in some participants, the resolution of the structural scans was different, and in the other participants with 0.8 mm anatomy the two voxel grids were probably not aligned

We clarified this part of preprocessing as follows (Lines 526-529):

“Next, we registered the ROIs with the predefined cortical depths to the EPI volume applying the coordinate mapping (with the voxel size resampled to the functional runs (0.8 isotropic)) obtained in the previous step (Figure 2C).”

22. 481: did the authors forgot to mention, or was there really no preprocessing of fMRI data (not even a high-pass filtering?). If so, perhaps it should be explicitly stated, because it is quite unusual.

We added a sentence stating that we did not apply any preprocessing to the functional data (Line 554).

23. 508 onwards: which software was used for statistical tests? Why FDR correction for multiple comparisons was used? This is not a typical multiple comparison correction method in the context of an ANOVA

We added the clarification in the Methods section (lines 578-589):

“We used repeated-measures analyses of variance (ANOVA) to test the main effect of Signal type (presented vs mentally rotated grating) in the trial and to test the interaction of Signal type and Cortical depths (deep and superficial vs. middle) (custom function `rmanova2` derived by A. Schurger (2005) from Keppel & Wickens, “Design and Analysis”, ch.18: <https://de.mathworks.com/matlabcentral/fileexchange/6874-two-way-repeated-measures-anova>). In cases where the assumption of sphericity was violated the p-values were corrected using a Huynh-Feldt correction (provided as an output of the same function). Significant interactions were followed up with paired-samples one-sided t-tests (`ttest` function in MATLAB) to analyze the effects in the assumed directions based on neuroanatomy and animal findings. To control for multiple comparisons across t-tests, we used FDR-corrections that assume

independent or positively correlated tests¹²: these corrections allow for maintaining a low false positive rate while providing reasonable power to find truly significant results.”

24. 588: “between subject SEM” – the authors mean the “standard error of the mean over subjects”?

Thanks, done.

Reviewer #2 (Remarks to the Author)

This is a very interesting study that makes a good contribution. I just have a few questions/concerns.

We thank the reviewer for their positive assessment of our work and provide responses to their questions below.

1. Focus on Mental Rotation?

The title of the paper suggests a focus on mentally rotated representations. This may lead the reader to expect a study looking at the representations during the rotation. However, it seems more likely that the current study is looking at the representation after the rotation or of a retrieved representation (imagined or internally generated). Therefore, I think it might be better to remove the focus on rotation from the title and elsewhere in the paper. Given that only a few orientations were ever presented, it seems possible that participants learned the responses and did not require mental rotation on each trial. I see that the RT data supports a mental rotation interpretation, but I don't find it entirely convincing. In any case, I don't think the assumption of mental rotation is needed to support the paper's conclusions, if the conclusion is that imagined, or internally generated representations are maintained in different layers of V1 than perceived representations.

Maybe I am missing a critical part of the argument, but it seems to me that the same thing would be predicted for a task that required participants to recall a given item from memory and maintain it for a later test as the current “mental rotation task”. If so, and the V1 representations are not specific to any sort of active mental rotation, then easing away from the focus on mental rotation may be best.

This is an interesting suggestion. In the manuscript, we indeed report the analysis of the item-specific contents which represent the result of mental rotation operation but not the process of reaching the end result. We realize that referring to cortical representations as “mental rotation signal” may lead to a confusion between the process and its end result. In the revised manuscript, we avoided this particular phrasing and used terms like “representation of mentally rotated contents” instead. We think that studying the representation of the outcome of rotation processes makes a clear contribution to the mental rotation literature, and we therefore prefer to keep the current framing. Our choice of words is further motivated by studies suggesting that there are neural and behavioral differences between representations of mentally rotated

contents and short-term retention (Christophel et al., 2016) or space-based imagery (Bainbridge et al., 2021), showing that these processes are not interchangeable.

We agree that there is a possibility that participants retrieved the task solution from memory rather than performing the rotation task. However, we consider this possibility unlikely, as we argue in the Methods section (lines 428-436):

“We cannot ultimately exclude the possibility that participants realized how many stimuli were shown overall and only retrieved the relevant orientation from memory rather than performing the rotation task properly. Such strategies are a typical problem in mental rotation studies using a fixed number of repeating stimuli. However, our behavioral data provides direct evidence against this scenario: the response time data cannot be accounted for by the retrieval of fixed orientations (or orientation) labels from memory. We would like to add that after the experiment, we asked each participant how many orientations they had to rotate and none of them reported the real number of orientations in the stimulus set. Therefore, we believe that our participants were genuinely performing the mental rotation task.”

2. Related to this is the timing of the classifier analysis. The authors note that “To test this hypothesis, we compared perception and mental rotation signals between the average of the outer layers and the middle layer, from 6-10 seconds after trial onset (i.e., 2 TRs), as done in previous studies”. It would be helpful to have this timing indicated on Figure 1A. My interpretation is that the analysis begins 4 seconds into the “mental rotation interval” and then covers the time when the test grating is presented. First, does the end of the “mental rotation interval” encompass mental rotation or the maintenance of the rotated or retrieved representation? Second, is it problematic to include the test grating pre-station time of the trial given that during this time a new stimulus is being perceived? Including a sentence or two about why this timing was chosen other than stating that this what was done in previous research may be helpful to the reader that is unfamiliar with the previous research.

We thank the reviewer for this suggestion. We now highlighted the analysis time window in Fig.1C (lines 60-70):

Figure 1. Experimental methods. A. On each trial, participants viewed a sample grating and then had 6 seconds to rotate it 60° (<, >) or 120° (<<, >>) to the left or to the right. After the mental rotation interval, participants had 2 seconds to report whether a probe grating was tilted clockwise or counterclockwise compared to the mentally rotated grating. **B.** We used a set of three stimuli, 15°, 75° and 135° oriented gratings. As a result of the mental rotation, each stimulus could be turned into one of the other two stimuli. For example, rotation of a 15° grating (red arrow) for 60° clockwise results in a 75° grating or rotation of a 135° grating (blue arrow) 120° counterclockwise results in a 15° grating. **C.** This panel shows classifiers' decisions in an example trial, in which a 15° grating was rotated into a 75° grating. We aggregated results across trials by counting how often classifiers predicted the presented orientation, the rotated orientation, and the unused orientation. The shaded area denotes the time interval chosen for the in-depth analysis (measurements at 8 and 10 seconds).

The mental rotation interval encompassed the time given to the participants to perform the mental rotation operation before probe grating onset. Thus, the end of the mental rotation interval is the end of the time limit for the rotation operation and a start of the task phase of the trial. We clarified this in the description of Figure 1A (line 61-64):

“On each trial, participants viewed a sample grating and then had 6 seconds to rotate it 60° (<, >) or 120° (<<, >>) to the left or to the right. After the mental rotation interval, participants had 2 seconds to report whether a probe grating was tilted clockwise or counterclockwise compared to the mentally rotated grating.”

In our study, we performed a cortical depth analysis at the time interval spanning measurements at 8 and 10 s after the rotation cue onset. In the initial version of the manuscript, we referred to this interval as 6 to 10 seconds, as every measurement reflects the signal dynamics over the interval of 2 s TR (6-8 s and 8-10 s)). This reference however might have caused a confusion; therefore, we adjusted the wording and the plots to represent specific time points exactly when the signal was acquired (2, 4, 6, 8 and 10 seconds after the trial onset, Supp. Figure 1 in the revised manuscript) and when the cortical depth analysis was performed (Figure 1).

The critical temporal window also included the time point right after the probe grating onset. Because of the temporal delay in BOLD signals, this time point is very unlikely to contain information about the probe grating. We have added a more detailed clarification for the analysis time window to the Methods description (lines 571-576):

“For the in-depth assessment of mental rotation contents, a critical time interval was chosen based on the previous studies (⁴⁻⁵ in the main references) where mentally rotated representations could be decoded in the period 8-12 seconds after the rotation cue. In our study, we included time interval 8-10 s after the rotation cue since the measurement at 12 s was likely to carry the representation of a probe grating (shown at 8 s), while the measurement at 10 s is too close to the presentation of the probe grating to be contaminated by it.”

3. Power

Why did the authors choose to collect data from 23 participants? Was a power analysis conducted?

We did not conduct a power analysis. We added a justification for our sample size to the Methods section (lines 367-368):

“We chose the number of subjects that was similar or exceeded the sample sizes of previous 7T studies investigating feedback signals with laminar separation ^{14,15,18,20,38}”

4. Localizer concerns?

It is explained that the localizer task consisted of presenting the gratings for 12 seconds and then also blocks of fixation for 15 seconds. What is the function of the fixation block? When is the data from localization task collected, only during the blocks when the gratings were presented or also during the fixation blocks? I am wondering if it matters if the localization task uses perception activation rather than internally-generated activation.

The purpose of the 15 seconds fixation blocks in the orientation localizer task was to provide a baseline interval during which the BOLD signal was returning to its state before visual stimulation. These intervals also provided short breaks for the subjects since the flicker was straining the eyes. The data was analyzed from the blocks when the gratings were presented. We reasoned that isolating perceptual activations during our localizer task is one of the most effective approaches to increase signal-to-noise ratio, as orientation-specific perceptual activations can be localized robustly in block designs within brief amounts of scan time. We were also guided by this widely used approach in studies of visual perception and working memory (Harrison & Tong, 2009; Bettencourt & Xu, 2016; Rademaker et al., 2019).

5. *Are there any concern about doing the localizer last, after the experimental runs? Could the experimental runs change how the orientations are perceived? Might there be more top down influence on perception based on the experience of encoding, rotating and retrieving the orientations in the experimental portion of the experiment. Would the authors predict anything different if the localizer was completed before the experimental task?*

We do not think that experimental runs affected the way the orientations were perceived in the localizer task. Rather, we chose to present the localizer at the end of the experiment to not alert participants to the fact that there were only three orientations, as now mentioned in the Methods section (lines 447-450):

“The orientation localizer task was performed last in the experiment to ensure that participants did not notice that the orientations shown to them during prolonged periods in the localizer task (12 s) are the same three orientations as the ones briefly presented in the beginning of each trial of the main experiment.”

6. Length of Experiment

Are there any concerns about fatigue in the experiment? The authors report that “The average time for completing the whole experiment was 75 minutes excluding anatomical scans.” The anatomical scan took 40 minutes. So, the whole experiment took 115 minutes, correct? Why not report the whole time instead of reporting the time minus the anatomical scan?

Our participants were experienced observers, many of which had previously participated in 7T experiments. We therefore don't think that fatigue was a big issue. As suggested, we now also report the whole experimental time (lines 490-491).

7. *What does the participant do during the anatomical scan?*

We clarify in the Methods section (line 408-409):

“During anatomical scan procedure participants were encouraged to rest and move as least as possible to reduce motion artefacts.”

Reviewer #3 (Remarks to the Author)

The Ms deals with an interesting issue and I think that it can contribute to better understand the role of V1 in perception and imagery processes.

Specifically, the study investigates if perceived and mentally transformed stimuli during a mental rotation processing can reflect a laminar-specific activation in V1 in response to visuo-spatial information. The findings report high activations in the middle layer of V1 for perceived stimuli, and activations involving the superficial and deep layers of V1 for imagined rotations. Moreover, the study takes advantage of high-resolution fMRI (7T) and well-designed cognitive paradigm/tasks. However, in its present form it suffers of a series of problems, which need to be addressed.

We thank the reviewer for the comments and suggestions from which the revised manuscript has strongly benefitted.

1. I am not sure if the sample size is in line with other investigations. However, it would be more appropriate to have a more objective criterion (power analysis), explained on the basis of test conditions and variables.

We did not conduct a power analysis and unfortunately cannot conduct it post-hoc after the study. We added a justification for our sample size to the Methods section (lines 367-368):

“We chose the number of subjects that was similar or exceeded the sample sizes of previous 7T studies investigating feedback signals with laminar separation^{14,15,18,20,38}.”

2. A concern regards the choice of degrees of mental rotation: only 3 angular positions with a high level of chance level (as also recognized by the authors) and the low number of trials. This undermines the methodological robustness of the experimental task used. This point needs to be deeply addressed.

Thanks for this comment. As far as we see, there are two principal concerns with having a low number of stimuli (and here: angular positions).

(1) Participants could retrieve fixed orientations from memory rather than actually rotating the shown gratings in their mind’s eye. This point also addresses the following comment of the reviewer: “4. Given the limited number of angular degrees and trials, I wonder if the authors can exclude that the participant did not use other types of visuospatial or memory strategies to perform the task.”

We added the following clarification regarding this point to the Methods section (lines 428-436):

"We cannot ultimately exclude the possibility that participants realized how many stimuli were shown overall and only retrieved the relevant orientation from memory rather than performing the rotation task properly. Such strategies are a typical problem in mental rotation studies using a fixed number of repeating stimuli. However, our behavioral data provides direct evidence against this scenario: the response time data cannot be accounted for by the retrieval of fixed orientations (or orientation) labels from memory. We would like to add that after the experiment, we asked each participant how many orientations they had to rotate and none of them reported the real number of orientations in the stimulus set. Therefore, we believe that our participants were genuinely performing the mental rotation task."

(2) Low number of stimuli might negatively impact ecological validity of the findings. This point also addresses a further comment of the reviewer: "*3. The use of more mental rotations would have been beneficial in the comparison with the long tradition of research on mental rotations (e.g. Zacks, 2008)*".

We address this point in the Methods section (Lines 414-418):

"We limited the number of possible stimuli compared to the training session to increase signal-to-noise ratio per each sample grating and to enable signal differentiation at the level of cortical depth bins. In future studies, the use of richer stimulus sets may provide insight to whether the same neural processes, which govern the mental rotation tasks, are also performed on everyday objects during our daily lives."

5. It would be interesting to check for lateralization of the mean signal of each hemisphere (e.g. Roberts et al 2003; Frings et al 2006 and others).

We thank the reviewer for this suggestion. The aforementioned studies showed gender-related differences in activity lateralization in parietal and hippocampal areas. Unfortunately, our current research was not specifically designed for testing this hypothesis. Among other reasons, the field of view used for scanning in our study covered only patches of early occipital cortex.

7. It's not clear to me if in the present paradigm the fixation cross could be used as an external reference point for completing the perceived or even imagined rotations. The Authors could address this concern.

In our paradigm the fixation cross was present on the screen throughout the trial. It is generally possible that participants somehow "anchored" the different grating orientations with respect to the fixation cross, but even if they did, we do not believe this invalidates our results, as the fixation cross remained constant throughout the experiment.

Reviewers' comments:

Reviewer #1 (Remarks to the Author):

The authors did an impressive work addressing my comments, especially the reason for comparing against the unused grating is convincing. I thank the authors for all the clarifications. I have only two very minor outstanding points that I think are still unclear:

Lines 526-529: which interpolation was used: nearest neighbour, trilinear, cubic, etc.?

Line 745: In the caption of supplementary figure 6 the authors say that the data was high-passed filtered – was this done only for the univariate analysis?

Reviewer #2 (Remarks to the Author):

The authors have mostly addressed my concerns. I just have a couple notes on how these concerns could be further addressed below.

Focus on representing mentally rotated contents rather than on the mental rotation process. There are still parts of the paper that make it sound like the study is examining the mental rotation process rather than the representation of mentally rotated content. Below are a couple sentences with the suggested edits to improve on this in bold and underlined.

“These results show how the representations of mental rotated contents is mediated by functionally distinct neural representations”

“Beyond the spatial separation of representations of perceived and mentally rotated contents, the laminar organization of feedforward and feedback information may also facilitate interactions between these signals.”

Power

The authors should consider the possibility that their study is underpowered to detect some effects. For example, is there a chance that a lack of power prevented detection of these effects? “No statistically significant difference between the deep and superficial layers was found when analyzing perception ($t_{22}=0.86$, $p=0.19$) or mental rotation signals ($t_{22}=0.6$, $p=0.27$).”

Reviewer #3 (Remarks to the Author):

Before suggesting the acceptance for this contribution, I still have some concerns about the study, particularly for power analysis.

Admitting that an a-priori (or a-posteriori) power analysis was not conducted does not absolve the authors from finding statistical consistency for the sample size.

Performing post-hoc tests with simple softwares (e.g. Pangea, Gpower..) could help to confirm or not the sample size. A further solution could be to compare the effect sizes with those of similar studies.

In general, this would help the authors to verify that the sample size was adequately powered to produce reliable and replicable results.

We thank all the reviewers for their positive assessment. We provide a structured reply to their remaining remarks below. The reviewers' comments are italicized, our replies are written as a plain text and citations from the manuscript are highlighted in grey.

Reviewer #1 (Remarks to the Author)

The authors did an impressive work addressing my comments, especially the reason for comparing against the unused grating is convincing. I thank the authors for all the clarifications. I have only two very minor outstanding points that I think are still unclear:

We are very happy that the reviewer found our clarifications convincing and we thank the reviewer for the further remarks.

Lines 526-529: which interpolation was used: nearest neighbour, trilinear, cubic, etc.?

We added this information to the same sentence (Lines 531-534):

“Next, we registered the ROIs with the predefined cortical depths to the EPI volume using nearest neighbor interpolation and applying the coordinate mapping (with the voxel size resampled to the functional runs (0.8 isotropic)) obtained in the previous step (Figure 2C).”

Line 745: In the caption of supplementary figure 6 the authors say that the data was high-passed filtered – was this done only for the univariate analysis?

We used the high-pass filtering for the univariate analysis only, as now mentioned in the figure caption (Lines 750-752):

“Prior to the analysis, the voxel responses both in the orientation localizer task and in the main experiment were high-pass filtered using spm12 (but not for the multivariate analysis (see Methods)).”

Reviewer #2 (Remarks to the Author)

The authors have mostly addressed my concerns. I just have a couple notes on how these concerns could be further addressed below.

Focus on representing mentally rotated contents rather than on the mental rotation process.

There are still parts of the paper that make it sound like the study is examining the mental rotation process rather than the representation of mentally rotated content. Below are a couple sentences with the suggested edits to improve on this in bold and underlined.

“These results show how the representations of mental rotated contents is mediated by functionally distinct neural representations”

“Beyond the spatial separation of representations of perceived and mentally rotated

contents, the laminar organization of feedforward and feedback information may also facilitate interactions between these signals.”

We thank the reviewer for pointing this out and we corrected the mentioned sentences accordingly:

(Lines 54-56):

“These results show how the perceived and mentally rotated contents are mediated by functionally distinct neural representations, ...”

(Lines 212-214):

“Beyond the spatial separation of representations of perceived and mentally rotated contents, the laminar organization of feedforward and feedback information may also facilitate interactions between these signals.”

Power

The authors should consider the possibility that their study is underpowered to detect some effects. For example, is there a chance that a lack of power prevented detection of these effects? “No statistically significant difference between the deep and superficial layers was found when analyzing perception ($t_{22}=0.86$, $p=0.19$) or mental rotation signals ($t_{22}=0.6$, $p=0.27$).”

We agree that there is a possibility that some effects in our study might not have reached significance due to insufficient statistical power. However, a-posteriori statistical power estimates have been argued to directly relate to the p-values established in the experiment. They therefore do not add any information over and above the p-values (Lakens, D. (2021). Sample Size Justification). We thus instead suggest an estimate of the effect size detectable in our experiment. With our sample size ($N=23$) and at statistical power 80%, a medium size effect is detectable in our study ($d=0.62$, paired-samples two-sided t-test). We added this information to the manuscript (Lines 371-373) .

Furthermore, we now explicitly acknowledge that some differences with smaller effect sizes could have been detected only with larger samples (Lines 153-156):

“No statistically significant difference between the deep and superficial layers was found when analyzing perception ($t_{22}=0.86$, $p=0.19$) or mental rotation signals ($t_{22}=0.6$, $p=0.27$); however, we acknowledge that these and other non-significant differences in our study might have reached significance with larger samples.”

Reviewer #3 (Remarks to the Author)

Before suggesting the acceptance for this contribution, I still have some concerns about the study, particularly for power analysis.

Admitting that an a-priori (or a-posteriori) power analysis was not conducted does not absolve the authors from finding statistical consistency for the sample size. Performing post-hoc tests with simple softwares (e.g. Pangea, Gpower..) could help to confirm or not the sample size. A further solution could be to compare the effect

sizes with those of similar studies.

In general, this would help the authors to verify that the sample size was adequately powered to produce reliable and replicable results.

We thank the reviewer for this remark.

We agree that there is a possibility that some effects in our study might not have reached significance due to insufficient statistical power. However, a-posteriori statistical power estimates have been argued to directly relate to the p-values established in the experiment. They therefore do not add any information over and above the p-values (Lakens, D. (2021). Sample Size Justification). We thus instead suggest an estimate of the effect size detectable in our experiment. With our sample size (N=23) and at statistical power 80%, a medium size effect is detectable in our study ($d=0.62$, paired-samples two-sided t-test). We added this information to the manuscript (Lines 371-373) .

Furthermore, we now explicitly acknowledge that some differences with smaller effect sizes could have been detected only with larger samples (Lines 153-156):

“No statistically significant difference between the deep and superficial layers was found when analyzing perception ($t_{22}=0.86$, $p=0.19$) or mental rotation signals ($t_{22}=0.6$, $p=0.27$); however, we acknowledge that these and other non-significant differences in our study might have reached significance with larger samples.”